# Contested Visions for Transformation—The Visions of the Green New Deal and the Politics of Technology Assessment, Responsible Research and Innovation, and Sustainability Research

**Christoph Schneider *** , **Niko Wilke and Andreas Lösch**

Institute for Technology Assessment and Systems Analysis, Karlsruhe Institute of Technology, 76131 Karlsruhe, Germany; niko.wilke9@kit.edu (N.W.); andreas.loesch@kit.edu (A.L.)
* Correspondence: christoph.schneider@kit.edu

**Abstract:** Societal transformations are contested. The goals and visions of transformations, as well as the means and strategies to achieve them, are born in political conflict and power constellations. Which transformations are seen as desirable and possible by democratic majorities changes throughout history. This is the political reality where research for transformations finds itself. Technology assessment (TA), responsible research and innovation (RRI), and sustainability research (SR) are a part of such contestations. They engage in envisioning, debating, analyzing, and evaluating different visions of and options for the future. In this article we turn to visions of the future as a key aspect of societal contestation and the shaping of interfaces between research for transformations and society. Based on the approach of vision assessment developed in TA, we situate TA, RRI, and SR within visions of research and social order. We argue that in these politicized times it is increasingly necessary to understand how research relates to larger visions of society and the contested nature of transformations. We turn to one of the major contemporary visions for societal transformation: The Green New Deal (GND). This vision imagines a large-scale transformation of society and the economy towards sustainability and justice and is currently debated in major political institutions and social movements. It presents an ongoing case of the "making of the future", which is highly relevant for TA, RRI, and SR. We show how this vision is creating new knowledge and social arrangements and how it is opening up new possibilities for transformational research. The article discusses the implications that a possible further impact of GND visions in politics may have for TA, RRI, and SR, and, relatedly, how expertise and insights from TA, RRI, and SR could significantly add to the GND debate.

**Keywords:** visions of the future; Green New Deal; politics; technology assessment; sustainability research; responsible research and innovation

## 1. Introduction

Societal transformations are contested, and transformative research participates in such contestations. The problems that societies face, as well as solutions and visions for the future for change, improvement, progress, and transformation, are shaped through debates, societal conflicts, and power constellations. The dominant visions of politics and the socio-economic order are very much about what is seen as "desirable" and possible by the majority of the population. Such landscapes also affect science, since they provide the main frames of interpretation of the world in public discourse [1,2]. This is precisely why technology assessment (TA), responsible research and innovation (RRI) and sustainability research (SR) constructively engage with different societal visions in the contested political domain. Writing from the perspective of TA, we argue that these transformative research have enabling conditions, which include the larger visions of societal change. Yet, they can

also have an impact in debating these visions. However, visions change over time, and so do the forms of social and political order that are legitimated or questioned through them.

Such change has emerged since the financial crisis of 2008. Western societies have gone through a phase of crises in the neoliberal era that was installed in the 1970s and 1980s. The neoliberal consensus, that was powerfully enforced by certain political actors (e.g., Ronald Reagan, Margaret Thatcher), national and global institutions (e.g., the International Monetary Fund), and other forces (e.g., global corporations), championed deregulated markets, global competition, and individualism. However, these ideas are, by now, in crisis. Last but not least, the need for a sustainability transformation challenges fundamental neoliberal assumptions that more competition and more growth would lead to a "better" future [3]. Following Antonio Gramsci, the late sociologist Zygmunt Bauman described the decade 2010–2020 as an "interregnum" in which "the old is dying and the new cannot be born", a phase of disorientation, contestation, and the search for new visions and power constellations that could lead to a new societal consensus [4].

A major vision in this interregnum is the so-called Green New Deal (GND). A vision supported both by social movements and political institutions that has recently reshaped the debate about climate politics and social justice. We could say that we are in the midst of the contested (possible) birth of a new socio-political order, which is why we chose to analyze the GND empirically. The GND imagines large-scale state investment in a green economy to meet the challenges of the climate crisis. This vision not only acclaims a new power arrangement between state and market, but also prominently features technological innovation, research, and sustainability. It, therefore, has important implications for TA, RRI, and SR. Furthermore, TA, RRI, and SR could critically engage with the GND approach and potentially provide relevant expertise to make GND debates more reflexive and complex. Yet, how can such a vision and its effects on the present be understood?

There is a growing academic literature on the GND. Key perspectives in these debates include the funding of the GND as a huge investment program [5,6], the GND as a just transition with workers at its core [7], democratic legitimacy of GND policies and measures, the eurocentrism of GND politics vs. the need for a global GND or decolonizing climate politics [8], as well as the question of economic growth and sustainability in relation to GND [9].

In our article we take a step back from such a direct assessment and critique of GND policies and ask why there is such a debate in the first place, i.e., how has this vision of a socio-economic-technological transformation come into being and gained such momentum? We approach the GND with the method of vision assessment, which is used in technology assessment and science and technology studies to examine the effects of visions on the shaping of new technologies. In fact, the article approaches the GND as a political innovation. We do not so much ask what the GND is, but rather, how it came about and how it is being "done" through visionary communication. In this way, we contribute to research on the GND by demonstrating the importance of the discourse of the GND to performatively shape GND realities. Secondly, we add to the literature in science and technology studies, technology assessment, and sustainability research by entwining contested political communication with perspectives on sociotechnical change. These tend to focus on innovation actors, technologies, and users rather than on the political landscapes that shape innovation.

In this article we utilize the vision assessment approach that was developed in TA to analyze and assess future visions of new technologies. Vision assessment reveals the dynamics and effects of visionary ideas in the present. The main empirical section of this article analyzes different visions of a GND and their implications for ongoing policy debates. We analyze these visions with a particular focus on science as an influential component of these visions, as well as its specific role in the transformations that a GND seeks to enable. We show that GND visions are strongly affecting the political landscape in Western societies through sketching a specific political-economic strategy towards a more sustainable economy and society. Finally, the article seeks to discuss the potential a sustained strength

of these visions could have for TA, RRI, and SR, and what possibilities there are for these fields to act strategically both with and through the visionary GND debates.

## 2. Theoretical and Empirical Background

This chapter discusses the methodology and approach of vision assessment as a way to analyze the role of visions of the future in the present. TA, RRI, and SR are then considered from this perspective to show the relationships of these fields to particular visions of social order. Finally, the chapter describes the empirical process for the analysis of GND visions.

### 2.1. Vision Assessment—Visions and Power

In the present, the future is in principle inaccessible. However, to enable future-oriented decisions and actions towards societal transformations, it is necessary to imagine the future. Therefore, we always find multiple, and often contradictory, imaginations of how the future could or should look and how future states of society can be achieved. We are often confronted with highly uncertain and "contested futures" [10], which are introduced by specific actors and often controversially debated in the different arenas of society. However, despite their uncertainty, such futures, which we call visions, shape processes in the present with impacts on shaping the future [11]. Such impacts of visions cannot be understood without reflecting on the social constellations of power. Defining which future is perceived as possible and desirable is a key question of power [12]. Political debates, decisions, and also research and innovation practices, are influenced by the far-reaching promises of science-policy visions, e.g., concerning the potential and impacts of new technologies for transformations in science and society, but also warnings about potential dystopian effects of future technologies for society and humankind.

Research, e.g., from the sociology of expectations [13], on hegemonic socio-technical imaginaries [1], or from vision assessment in TA [14,15], has provided many insights into the mediality and performativity of future expectations, visions, and imaginaries in current processes of change in our "TechnoScienceSociety" [16]. Such visions serve as a resource in political economies because creating a focus on specific visions in socio-political debates and practices can be seen as capacity to increase value or mobilize support and, correspondingly, to stabilize or to transform existing power constellations [17]. While this research has shown the effects of visions on social reality, research on the complexities of sociotechnical transformations has pointed out the multi-dimensional nature of such large processes of change. Sustainability transformations or infrastructure change involve messy, layered, and non-nested processes wherein different actors pursue strategic agendas, existing sociotechnical systems have momentum and path-dependency, and cultural norms and practices are enacted in everyday life [18]. This means that no single vision or social actor pursuing such a vision can decisively shape these transformations. However, within such complexity of transformations, politics and states can play important roles along with visions that guide policy-making [19]. Thus, visions are an important aspect to study within the complex societal contestation of transformations. However, because of the complex nature of such change, studying visions cannot claim that the desires within these visions are realistic or feasible. Instead, studying visions helps to understand how complex social realities are shaped and contested through the effects of visionary communication.

The relationship between science and politics has a long history and has been fundamental for modern knowledge societies since the mid-20th century. For politics, its relationship to science is important to obtain robust scientific expertise to legitimate political decisions; science is also needed to create imagined futures, scenarios, and visions, which could be used to orientate decisions on, for example, the funding or regulation of future technologies, which should contribute to sociotechnical innovations and transformations. The role of science in this relationship is two-sided: on the one hand, science has the role of a producer of visions for politics, but on the other, science is also the responsible addressee for politics in order to contribute to the fulfillment of its own visions. This can be characterized as a simultaneous "scientification of politics" and "politicisation of

science" [20]. Following this perspective, science is not autonomous from political developments, ideologies, and dominant ideas that define politico-economic orders. Science is embedded in such political, cultural, and economic landscapes of societies and is, thus, strongly affected by changes in the dominant political ideas of the day [21]. Reflecting on this, it is obvious that power dimensions of visions can only be understood, and maybe changed, by pondering on the specific arrangements between science, politics, and the economy, in which certain visions are created, deliberated, and distributed. Such visions always construct specific constellations of actors, which should be addressed, included in communications, and activated to contribute to the fulfillment of the visionary goals.

We can analyze the impact of visions by applying the vision assessment approach of TA [22]. Vision assessment tries to analyze processes of visionary communication and their effects in science, politics, economy, and other spheres of society. It considers attributions of meaning to technologies that do not yet exist through visions in current debates (e.g., in research policy, science, or the mass media), with the goal of evaluating epistemic and normative assumptions underlying visionary discourses, and of enabling responsible monitoring (or modulation) of the discursive use of visions in scientific, policy, and public debates [23]. As a practice-oriented approach, it tries to analyze and evaluate not only the contents of visionary motives or narratives and their underlying epistemic and normative assumptions, but also the functions and impacts of the visions in practical use contexts of current research, development, innovation, and transformation processes, such as negotiation practices, laboratory practices, and interactions with stakeholders [24].

From the point of view of vision assessment, TA, RRI, and SR can be located within science-politics arrangements and their related visions of society and strategic agendas for shaping the future.

TA was first institutionalized close to parliaments. The vision that informed this institutionalization was that of enabling a more reflexive representative democracy in the light of technological change. In this idea of order, TA should provide neutral technology expertise used to inform value-based, political decision-making, as opposed to interested technological expertise in industry [25]. Of course, this dichotomy has been challenged by science studies, yet it remains a legitimizing idea of this arrangement, which is, however, currently highly controversial—also with the necessity of transformations looming ever larger over Western societies [26].

The vision that RRI strives for is that of multi-stakeholder governance, through which various actors respond to one another in innovation processes to shape innovations through the lens of societal values, such as sustainability, inclusion, or openness. RRI seeks to activate state, industry, science, and civil society actors to become co-responsible for innovation. Classical market mechanisms should be complemented by diverse networks of stakeholders and their values in order to shape innovation. We could say that in its European version, RRI is invested in a larger vision of a more responsible stakeholder capitalism that would seek to balance many values besides profit maximization [27,28].

With the idea of Sustainable Development, the United Nations define a clear vision and a political project for transformation, which, since 2015, is typically articulated through the 17 Sustainable Development goals. This forms the normative center of much SR. Besides this rather concrete institutionalization, sustainability is a central concept in many societies, which is, however, strongly contested. Different strategies, definitions, actors, and power constellations interpret the meaning of sustainability differently [29].

From this brief comparison, a striking similarity between TA, RRI, and SR becomes apparent: they are "inside outsiders", positioned both within and beyond political arrangements and institutions. They are simultaneously close to a particular arrangement of political power, yet their mission is to provide input for certain forms of transformations. These research fields are related to certain visions of social order and the desired transformations that result from it. Yet, it is precisely this double role that makes the three fields so dependent on political opportunity structures, which affect what is seen as possible and desirable in political institutions or in the wider political discursive landscape.

The remainder of this article turns to an empirical analysis of a specific vision of the future: the GND. This vision is not yet as strongly institutionalized as those just discussed, yet it is restructuring political discourses, and may enable pathways and openings for research and expertise in TA, RRI, and SR. In the empirical analysis we highlight such possible intersections of these research fields and the effects of the GND visions.

### 2.2. Empirical Analysis: Discourse Analysis

To analyze the positioning, function, and role of science within GND conceptions, we conducted a discourse analysis, drawing on key publications campaigning for a GND that have gained public relevance or serve as orientation for other GND documents [30]. In addition to recent books by public intellectuals on the GND, we also considered the 2007 newspaper articles by Thomas L. Friedman in which he introduced the term Green New Deal. Additionally, two Executive Orders issued by U.S. President Joe Biden, as well as the Bill introduced to Congress by Alexandria Ocasio-Cortez and Ed Markey calling for implementation of the GND, were included in the analysis. Finally, the European discourse was considered, including the European Commission's statement on the Green Deal for Europe and the report of the Green New Deal Group in the UK, as well as the report by the Green New Deal for Europe Campaign. Table 1 gives an overview of the source material. The selection of these texts is based on their position within the GND as political discourse. The three documents from 2007 and 2008 defined the name and the idea. The other documents are either political texts from within political institutions or texts by intellectuals who engage in the political process and have advised politicians or political movements. In doing so, we have tried to select texts that are at the center of GND policy debates.

The discourse analysis looked for key patterns of argumentation concerning the role and importance of science and its innovations in the divergent GND conceptions. The identification of these passages in the various texts was facilitated by a keyword search that included the terms: 'Science/Scientist, Technology, Innovation, Social Sciences, Humanities, and University'. The sections including these terms were analyzed in more detail concerning their key messages. Furthermore, one of the authors was a scientific consultant in the writing of the Green New Deal for Europe report, from which we draw insights from "behind the scenes". This entailed writing a section of a chapter on digital technologies in a GND and accompanying the coordination process that sought to involve many political actors from the progressive spectrum across Europe in the writing of the report.

The analytical strategy was to link key messages found in the texts with the actions of GND promoters and political events, i.e., what was the political landscape like when the texts were published or used and what was being done by GND promoters to influence this landscape. It was important to relate language and discourse to political practices and constellations to assess the actual impact of GND visions. The conceptual framework of vision assessment helps to focus on the practical functions of visions: serving as temporal interfaces and normative forces, enabling communication, and assisting coordination [15,24]. These practical functions and their effects are made empirically visible in our case study on the science and visions of a GND. The following section presents the analysis of the socio-epistemic practices of GND visions.

**Table 1.** Key publications of GND.

| Title | Author | Year | Type of Document |
|---|---|---|---|
| A Warning from the Garden | Thomas L. Friedman | 2007 | Newspaper article |
| The Power of Green | Thomas L. Friedman | 2007 | Newspaper article |
| A Green New Deal—joined-up policies to solve the triple crunch of the credit crisis, climate change, and high oil prices | Green New Deal Group/New Economics Foundation | 2008 | Report |
| European Green Deal | EU Commission | 2019 | Report |
| The Green New Deal: Why the fossil fuel civilization will collapse by 2028, and the bold economic plan to save life on earth | Jeremy Rifkin | 2019 | Monograph |
| Green New Deal For Europe: A Blueprint for Europe's Just Transition | Democracy in Europe Movement 2025 | 2019 | Report |
| A Planet to Win: Why We Need a Green New Deal | Kate Aronoff; Alyssa Battistoni; Daniel Cohen; Thea Riofrancos; Naomi Klein | 2019 | Monograph |
| On Fire: The Burning Case for a Green New Deal | Naomi Klein | 2019 | Monograph |
| The Case for The Green New Deal | Ann Pettifor | 2019 | Monograph |
| A Message from the Future | Alexandria Ocasio-Cortez; Naomi Klein | 2019 | Video |
| Executive Order on Tackling the Climate Crisis at Home and Abroad | The White House | 2021 | Executive Order |
| Resolution: Recognizing the duty of the Federal Government to create a Green New Deal | Alexandria Ocasio-Cortez; Ed Markey et al. | 2021 | Congress Resolution |

## 3. Results: Collective Envisioning of Green New Deal(s)

This chapter starts with a brief history of GND visions and then draws on the analytical dimensions of vision assessment to discuss how these visions create an interface between present and future, how they activate a normative force, how they enable communication, and, finally, how they coordinate different actors.

### 3.1. A Brief History of the Green New Deal(s)

The GND can be seen to start with the original "New Deal" that was implemented by Franklin D. Roosevelt in the U.S. after the world economic crisis in the 1930s. The New Deal was a vast government-led program to restructure the U.S. economy, invest in workplaces, and regulate finance. The New Deal tried to rebalance economic power in favor of workers and society and is widely regarded as a "progressive" reform. Its effects lasted until the 1980s.

The notion of a "Green New Deal" was first published in 2007 in a New York Times article that argued for "green" to become part of "America's DNA" in order to bolster its

geopolitical power as a leader in green energy [31]. Later that year, the financial crisis hit and sent the world into economic turmoil. This is the background to the meetings of a group of journalists, politicians, unorthodox economists, and environmental activists in the UK in 2007, which resulted in the report "A Green New Deal. Joined-up policies to solve the triple crunch of the credit crisis, climate change and high oil prices" [32]. The report suggests that: (a) the state should finance the transition to a renewable energy economy, create jobs, and regulate private finance and (b) the state should rebuild the economy, but put economic development on a sustainable path. Some politicians, political forums, and parties adopted the idea of a GND, inspired by this report [33].

Another crucial time for the GND visions was 2018–2020. In 2018, the climate justice movement transformed into a global youth movement, inspired by Greta Thunberg's school strikes. Several books by public intellectuals appeared that feature this notion and add ideas to the vision [34–38]. The unorthodox economist Ann Pettifor co-wrote the Green New Deal Report in the UK in 2008. She had also been consulting the "Democracy in Europe Movement 2025". This movement framed its policy proposals as a "Green New Deal for Europe" [39] in the run-up to the European Parliament elections in 2019. Public intellectual Naomi Klein is widely read in the U.S., where the "sunrise movement" has been mobilizing since 2017 for a Green New Deal including good jobs and a sustainable future [40]. U.S. congresswoman Alexandria Ocasio-Cortez has supported the movement and, together with Naomi Klein, released a video called "A message from the future", which tells an imagined story of a successful GND [41].

Major shifts towards GND policies have taken place in U.S. politics. A resolution for a GND to U.S. Congress was published in early 2019. Bernie Sanders campaigned on a GND platform for presidential candidate [42]. In early 2021, Joe Biden announced major policies on climate protection and infrastructural transformation, including key GND ideas [43]. Naomi Klein and Alexandria Ocasio-Cortez celebrated this as a victory for the GND vision in public statements. In Europe, similar shifts seem under way. In 2018 and 2019, Europe was the key site for the Fridays For Future Movement and the 2019 EU Parliament elections saw a huge rise in votes for green parties, resulting in the European Commission adopting a "Green Deal", focusing public investment in innovative and sustainable technologies.

We might say that the decisive shift around 2018 concerning the GND visions was that social movements effectively adopted them for extra-parliamentary democratic politics. This turned GND visions from a technocratic policy framework into a force for social movement organization and created a new political constellation. Figure 1 gives on overview of this history.

### 3.2. Creating an Interface between Present and Future

The first function that we focus on is that of a visionary discourse creating an imaginary interface between present and future. Visions provide a way of seeing the world, an interpretation of what the problems and possibilities of the present are, and sketch a picture of a desirable future and ways to get there. This is achieved through visionary narratives that tell a story of how to get into a specific desired future [44].

The key narrative at the heart of GND visions could be summed up in the following way, which is is not a direct quote but a condensation of our empirical findings: "Climate science tells us that it is urgent to act to prevent climate catastrophe. To do this, the state needs to restructure the economy from 'fossil reliant' to 'green'. This green economy must protect the climate and be just and sustainable to provide a safer and happier life for all humans. Such a massive transformation in a short time span is possible – look at the original New Deal. We (the people who want a GND) can achieve this by democratically winning state power and demanding a GND".

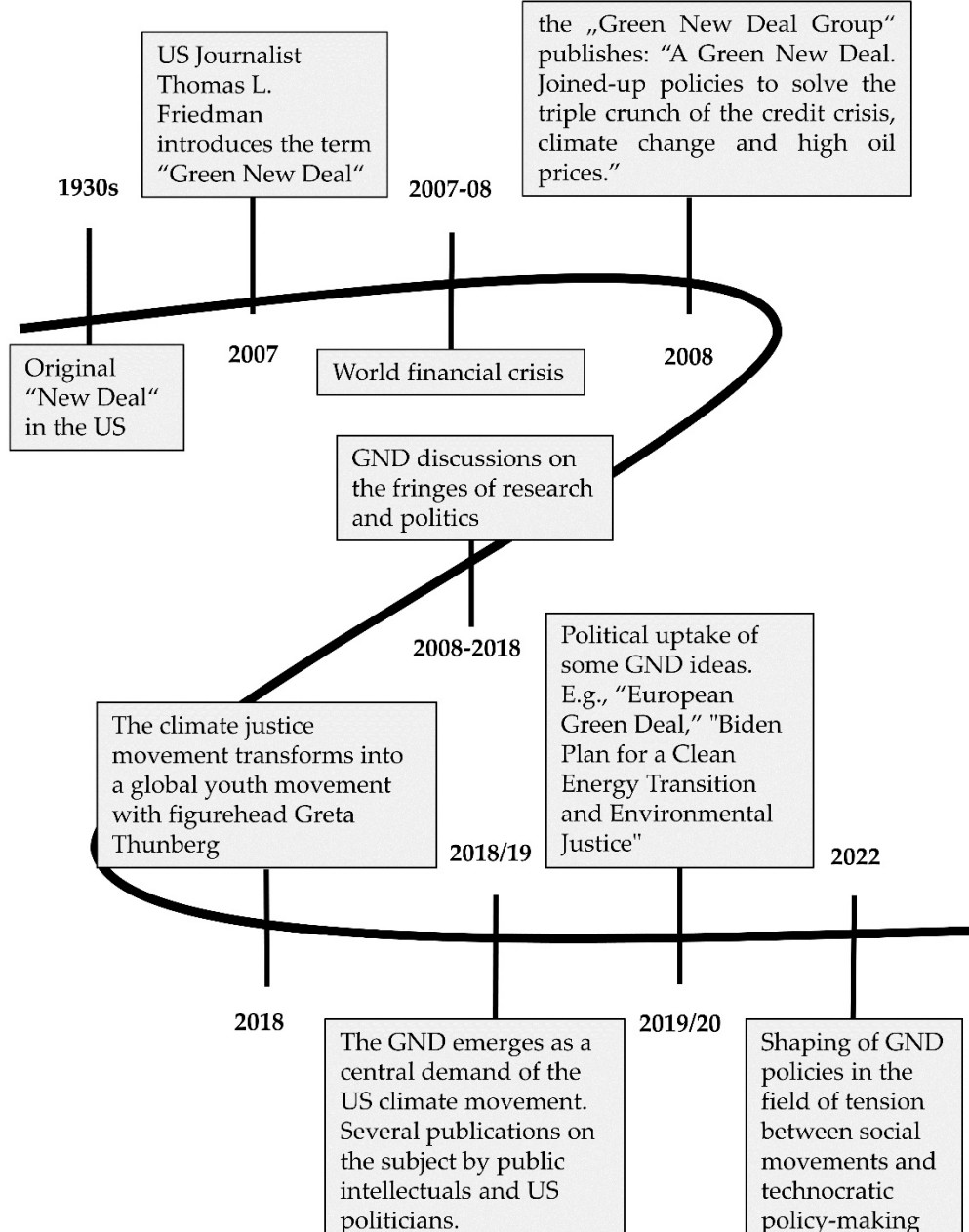

**Figure 1.** Timeline of central GND events.

Overall, there is relatively little explicit reference to science in GND texts. However, certain key ideas about science in society are identifiable. Already in the first publications on a GND, its advocates argue for an authoritative understanding of science, as an institution that must be trusted and that provides guidance. GND visions assign science a central role as the foundation of policy—a classic view born in the enlightenment. This should be seen against the backdrop of massive and organized climate denialism in the political arena [45]. The GND is, thus, a political vision that seeks to strengthen the role of science in politics—correspondingly, it would nourish the legitimation of TA, RRI, and SR.

Crucially, however, the GND visions imagine a vast and systemic transformation that would involve state, politics, the economy, and civil society, as well as science and research to be reorganized on a grand scale. In the GND visions, this mainly translates into state investment in green jobs and infrastructure.

The focus on the urgent and fast transition translates into a specific role for research and innovation. The GND advocates new technologies, such as renewable energy and

electric mobility. However, the visions claim that it is necessary to use existing technologies, while at the same time funding research for climate protection [36]. Thus, science, research, and innovation are already imaginatively embedded as just one element in systemic transformation. The same is true for technological innovation, which is seen positively in the visions. However, a central aspect in the GND visions is that innovation *processes* must be organized differently than they currently are: the state should invest in innovation, and give it a direction towards sustainability; some GND texts explicitly advocate global open source sharing of new technologies, instead of private patents [36] (p. 29). Such a systemic framing of change is also an aspect that TA, RRI, and SR share—with a focus on sociotechnical and socioecological change. So again, there could be overlap and mutual reinforcement between the research fields and the visions and projects of the GND. Paradoxically, social science is almost absent in the GND texts. However, some of the aforementioned key visionaries, such as Naomi Klein, Ann Pettifor, or Jeremy Rifkin, are in fact critical social scientists and public intellectuals. Through their involvement and advocacy for a GND, they communicate an understanding of social science that should be involved in societal transformations, inspire social change, and foster political activism. Implicitly, the GND visions are, thus, also about the position of critical social scientists in political networks and social movements.

### 3.3. The Normative Force of the GND Vision

Visions have a normative force that compels certain groups to take action, either to try to realize a vision or to try to prevent it. In this normative dimension, the focus is on what values are being debated through the vision and how this relates to different actors and groups in society. Envisioning new and "better" realities is a key aspect of political framing through which values are articulated. Values are variously interpreted in different groups and societies, with progressive, liberal, and conservative groups diverging strongly on particular values and related worldviews [46]. This is why the normative force of visions is a key political dimension of visionary communication.

Key normative frames debated through GND visions can be described as: a fast and systemic transformation, the democratic state taking a key role in managing change, enabling sustainability and social justice, and trust in science and the possibility of progress. These are progressive values voiced by actors on the progressive political spectrum [46,47]. GND visions can be identified as factional and politicized—visions that may attract people with progressive politics and potentially repel people with economically liberal or conservative worldviews. In this way, the GND visions differ from, for example, the SDGs, which are based on the universalist rhetoric of the UN and are agreed upon by all states in the UN as goals that should be considered desirable everywhere. Furthermore, and this is key to the strategy of the GND vision, it is used to mobilize supporters.

At this point, one can distinguish two strands of GND visions: a *technocratic* version, which is more dominant within political institutions, and a *social justice* version, which prevails amongst social movements. Since GND visions have entered mainstream political institutions to some degree (U.S. congress, EU commission and parliament, some political parties) there is an inherent normative tension in the constellation of actors. The first major political successes in these institutions were paired with a more moderate interpretation of GND ideas with less radical and less systemic approaches. However, energized social movements are advocating for "radical" GND visions that question existing distributions of power, inequality, and the current design of the institutions themselves. Such a constellation could be challenging for TA and RRI as both are strongly embedded in technocratic institutional politics, whereas SR has been engaged with various social movements for sustainability and technocratic policymaking.

The more moderate and technocratic interpretations focus on reducing carbon dioxide through new technologies and "green" industries. The more radical versions argue that a new social contract is needed, which complements carbon efficiency with social justice measures [48] and more democracy. Working time should be reduced, "green" workers

be organized in unions, and participatory councils should decide about investments [36]. This also has implications for the role of science. In a technocratic GND that focuses on eco-efficiency, science could be incorporated in expert-centric state planning. In a social justice GND, many more stakeholders would necessarily be included in transition planning. Citizens, social movements, civil society actors, and more, would need to be given possibilities to cooperate with the state in the planning process. This is clearly voiced as a goal in the visionary GND texts [37,39]. For science, research, and innovation this would then entail a more transdisciplinary and cooperative method of operation [37] (p. 271)—potentially similar to that already practiced in RRI and SR.

### 3.4. Enabling Communication

A further crucial function of visions is to enable communication between different actors. Visions serve as a means of communication; they are shared points of reference in a discourse. Through enabling communication processes, visions are generative and set an imaginative framework that is open to contestation and heterogeneous visioning.

A key aspect here is communication about defining the vision itself. From setting the initial visionary framework of a GND in 2008 until today, new aspects and ideas have been added to the GND visions. For some actors, this was key to their GND strategy: to enable a debate between different groups and political traditions. Moreover, this communication about the vision also takes place in the public sphere. It is a key aspect of the political nature of a vision if it is able to structure public debate about the future. One indication of the power of the GND visions in public debate is that there is also massive criticism of the idea from some commentators and groups [46]. Besides the political domains of activism and public discourse, GND visions are also debated in scientific discourse. In addition to the engaged scientists involved in the creation of the key visionary texts, there has also been a growing scientific engagement with the GND ideas in various fields. An important debate is whether or not the GND should boost economic growth and which forms of growth are fostered by its policy measures. This mirrors key debates in SR [9].

Once debated in politics, GND visions become ways to reinterpret policy areas. An example here is the EU's Green Deal strategy, which connects various strands of EU activity that existed before the new wording was used. However, such an interpretation of what is can also become a means of contestation. The Green New Deal for Europe campaign, for example, criticized the EU Green Deal for "greenwashing" old and existing EU structures and measures by re-branding them with GND language [49]. Such reinterpretation and shaping of policy could also be GND pathways for TA and RRI with their close connections to policymaking.

Through their public presence, GND visions have become an opportunity to engage with the present and future through their lens. What are the problems of the present and what are better ways into the future? In this way, GND visions fulfill an important democratic function, enabling societies to debate alternatives for change. Alternative visions of the future fill the political space at the heart of democracy [2].

### 3.5. Coordination of Heterogeneous Practices

Visions enable different actors to see their practices as being under the "same umbrella"; they coordinate heterogeneous practices. Typically, this does not take place in a deterministic sense of a controlled plan that is followed, but by providing an interpretative scheme by which different practices and their meanings can converge, and through which a sense of collective action and shared purpose can emerge among different actors.

The texts of the GND vision imagine a coordinated effort by a variety of stakeholders who should work together. The key decision-makers in this vision are responsible, reflective, proactive policymakers who lead through reform programs, taxes, stimulus spending, and legislation, as well as the scientific community that guides policymakers with expertise and new technologies. Moreover, the importance of innovation is emphasized. Green investments should guide innovations and create jobs to attract private companies to

follow the vision. An engaged, empowered civil society, whose perspectives and specific expertise should be reflected in the policy output, is also imagined as an important actor. Especially disadvantaged population groups, as well as professional groups (e.g., coal industry or fishery) and activists, should be heard [43]. However, the degree of integration, participation, and weight of input from these groups diverges across GND conceptions. In technocratic versions a rather narrow, conservative understanding of science prevails. Institutes and universities only hold an advisory role. However, universities and other scientific institutions are relevant when it comes to preparing and educating the new "green" workforce [34]. GND visions with a focus on social justice pay more attention to giving all stakeholders, especially disadvantaged groups and social movements, a seat at the table to identify problems and solutions. In their role as inside outsiders, TA, RRI, and SR could enable such interfaces between different groups and bring their expertise in collective knowledge creation to the forefront.

There is much communication that assembles different actors and provides them with a shared meaning. This has been important in the process of creating the vision. As early as 2008, the GND report expressed that the vision aims to connect the concern for ecology in "green" politics with the concern of justice and good jobs in "red" (e.g., social democratic) political traditions [32]. The GND visions aim to reconcile the need for climate protection and the need for inclusive and equal participation in the economy. This coordination has taken place through the language of the GND discourse and through various activities, forums, and organizations that try to bring actors from different spectrums together. The GND can function as a shared narrative for heterogeneous actors and, for example, give politically progressive groups a common thrust. It is possible to discuss diverse notions, ideas, and visions under the label of the GND.

Politically, a key coordination effect that was enabled by GND visions is the friction between political institutions and social movements. Whereas some institutions have adopted certain GND ideas, the boldest and most far-reaching visions are being voiced by activists and social movements, who celebrate the success of GND visions, but also call for more ambition in policymaking. Kate Aronoff, journalist and GND activist, notes that successful GND campaigning would need both technocratic state management based on existing institutions, and compelling and engaging social movement politics aimed at democratizing these institutions [50]. This theory of social change—a strong civil society pushing the state and demanding a "new deal"—is communicated as part of the vision and articulated as a strategy, this reinforces the idea that citizens must be activated to fight for a GND.

We have discussed above how critical and public intellectuals have been key facilitators of the GND visions, and it is obvious how these visions and the debates they are engendering create spaces for these intellectuals to further engage with the media, policymakers, and others. So far, however, science and the scientific system in general are not strongly affected by GND visions on a larger scale. This could change with initiatives such as the EU Green Deal affecting science funding more strongly in the future. It is, nevertheless, an ongoing question how GND visions will become institutionalized and shape transformative realities. In the following section, we discuss what these results mean for TA, RRI, and SR.

## 4. Discussion: Research with and for a GND

Our empirical analysis shows how visions of a GND are already an effective force in debates and policies for societal transformations. GND visions occupy central spaces in the contest of political visions. And, although their further influence is uncertain, there are important implications for TA, RRI, and SR, which we discuss together with the results in this section. Crucially, the GND visions could significantly shift the political landscape and become a central narrative of change in the coming years. As we have pointed out, GND visions could more strongly legitimate transformative research fields and, thus, support the

agendas of TA, RRI, and SR. We discuss how the three research fields could engage with the visions right now.

There are certain limitations to the empirical analysis since we are dealing with ongoing, contested, complex and uncertain processes of envisioning a societal transformation. It is too early to say whether there will be a major transformation similar to an envisioned GND. However, for transformative research approaches in general and for the vision assessment approach that we have followed here, being in the midst of ongoing changes is not unusual. Another limitation is the empirical basis of our analysis: we had to rely largely on publications by the visionaries of the GND themselves, who, of course, have a biased view of their own practices and visions. This was balanced through drawing on media publications and social theory. Further empirical research, e.g., based on interviews with observers and stakeholders of the GND vision process, is still strongly recommended. Furthermore, critical literature points towards certain challenges of a GND, which should be further researched. Some critical dimensions include the question of financing a GND, the euro- and anglocentric approach to the global sustainability crisis, the role of the state in shaping transformations and the question of capacity building to democratically enable such state intervention, the inclusion of workers and marginalized groups in GND policy making, and the question whether stimulating growth through GND policies is conducive to sustainability at all. None of these challenges can be accomplished simply by the popularity of a visionary narrative. Yet, as our analysis has shown, the GND visions induce processes of change and strategic action in various fields and contexts. Visions can enable and support societal experimentation [51].

Despite these limitations, the social force of GND visions is evident. What possibilities do these ongoing changes to the (envisioned) landscapes of societal transformation entail for TA, RRI, and SR? How could TA, RRI, and SR add their expertise to the ongoing collective visioning of GND(s)? In our understanding of the role of GND visions, this could mean neutral evaluation of GND proposals, but in terms of transformative research approaches, it would mean actively advocating for and promoting GND visions.

First, we could say that the processes surrounding the GND visions bring many aspects to the forefront of political contestation that overlap with the agendas of TA, RRI, and SR: e.g., more reflexive politics and a civil society that engages with grand challenges, new technologies, and sociotechnical change (a goal of TA), the responsible direction of innovation based on values of sustainability and inclusion (a goal of RRI), and a systemic transformation towards a sustainable economy (a goal of SR).

Second, as we have shown, visions of a GND are a generative framework to ask questions and create new knowledge. Scientific debates dedicated to key questions and tensions of a GND exist, and TA, RRI, and SR researchers could engage directly in these debates with their expertise. Given the reactive approach that is dominant in TA, this would involve that GND ideas become part of actual policies and research missions. However, RRI and SR could take advantage of the engaged and normative traditions the fields entail and contribute to actively setting GND agendas through scientific debate and advice. RRI has already operationalized different ways to include values of sustainability and inclusion in innovation processes in collaboration with research, citizens, and innovation actors. SR has been debating sustainability and different pathways and strategies to achieve it, which sets the reference for the key goal of GND visions: a sustainable economy. Furthermore, proponents of GND visions could engage with the three research fields and their expertise. There are clearly intersections where mutual learning, constructive critique, and capacity building could be possible.

Third, the popularity of the GND visions points towards certain opportunities for transformation. TA, RRI, and SR, as fields in the midst of societal transformations, should listen closely to these shifts and engage with and through them in strategic capacity-building, readying the fields more strongly for major shifts that may also affect research. That state and government planning seem to be returning on a major scale, points back to the founding years of TA. Since then, TA has seen it as a major mission to show the

ambivalences and complexities of technological progress. A time might now be dawning when it is no longer enough to think more reflexively about such issues, but to be able to give pragmatic and effective advice to governments, state actors, and science organizations on planning suitable to the contemporary complexity [12]. The speed of innovation and transformation that is imagined in GND visions, and in line with demands by climate science, would mean a massive demand for reflexive and responsive innovation processes. RRI researchers could get ready to think big and operationalize ways to scale RRI approaches. The major goal—to deliver a more sustainable economy and society through GND measures—highlights the need for societal, real-time co-evaluation capacities of the sustainability effects of GND measures. This would need to involve SR researchers and other stakeholders in creating, shaping, and engaging in new knowledge infrastructures. The transdisciplinary traditions in SR could prove central for this.

Fourth, there is a major challenge of technocratic dominance in GND-inspired policy approaches, with experts and state agencies leading in a top-down manner. As we have shown, this is contested between institutions and social movements, and it is ultimately a question of the changing nature of democratic power. However, science's organization and hierarchies also come into play—and these have not been significantly addressed in GND discourse to date. Therefore, the debates in TA, RRI, and SR for participatory processes, democratizing expertise, and opening science, could be significant elements of a more democratic GND.

## 5. Conclusions

In this article we have analyzed the making of the visions of a Green New Deal and the effects of such visionary discourse on constituting a political landscape of the GND. Writing from the perspective of technology assessment we have analyzed these visions similar to visions of new technologies and asked how these visions enable a process of political innovation (instead of technological innovation). By now, GND visions and policies have entered political institutions and social movements and the question is less whether there will be a GND and more what kind of GND policies are created. With the focus on building state capacities for shaping transformations, including a stronger role of science in policy making as well as the combination of social justice and environmental perspectives, GND ideas are an important point of engagement for science on sustainability, technology assessment, and responsible innovation, as well as other fields. The terrain of an uncertain, yet emerging, future constitutes a key domain for transformative research fields. Here, central opportunity structures and debates about transformation are defined. We live through politically volatile times and things might look differently in the near future. Still, as we have already discussed, visions not only require analytical scrutiny, but can enable researchers to engage in visionary communication. A certain limitation of the article and strategic action focusing on visions of the future is that political power and contestations are highly mediated by other aspects: class, identities, institutions, geopolitics, political systems, etc., are part of shaping transformations. However, the starting point of democratic transformations is often visions that are debated in public. The public sphere is where TA, RRI, and SR and other transformative research fields could intervene more strongly, reflexively, and, indeed, in a more visionary way. This could help societal actors to envision a more democratic shaping of technological change, value-driven innovation processes, and sustainable development—and possibly contribute to a future with a GND.

**Author Contributions:** Conceptualization, A.L, C.S. and N.W.; methodology, A.L. and C.S.; investigation, C.S. and N.W.; data curation, N.W.; writing—original draft preparation, A.L., C.S. and N.W. All authors have read and agreed to the published version of the manuscript.

**Funding:** This research received no external funding.

**Data Availability Statement:** All data are freely accessible, as they are all taken from publicly available documents.

**Acknowledgments:** We'd like to thank our colleagues who have provided feedback on an oral presentation of the article's idea and the manuscript, as well as the reviewers for their constructive feedback.

**Conflicts of Interest:** The authors declare no conflict of interest.

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
