# Peer review of "Contested Visions for Transformation—The Visions of the Green New Deal and the Politics of Technology Assessment, Responsible Research and Innovation, and Sustainability Research"

_sustainability, doi:10.3390/su14031505_

Round 1

Reviewer 1 Report

Exploration and analysis of the GND, using a visioning approach. The ascendency of the GND as a concept and policy makes this a valid and interesting question to address and explore.

Much of the Materials and Methods section reads like a literature review, and indeed, I think it is. It is not a big point, but after the introduction, I was left thinking that it was quite light and I did not see within which debates you were situating your article. The introduction covers the material well, but does not really engage with any existing empirically/ policy orientated GND literature – a few more references might be helpful. In addition, you could retitle the Materials and Methods section as ‘Literature Review’ until section 2.2, which is where the methods starts.

Referencing some additional critical literature could also strengthen the section. For instance, recent critical articles I find include,

Galvin, R., & Healy, N. (2020). The Green New Deal in the United States: What it is and how to pay for it. Energy Research & Social Science, 67, 101529.

Long, T. B., & Blok, V. (2021). Niche level investment challenges for European Green Deal financing in Europe: lessons from and for the agri-food climate transition. Humanities and Social Sciences Communications, 8(1), 1-9.

Rowe, J. K. (2020). The Green New Deal, Decolonization, And/as Ecocritique. New Political Science, 42(4), 624-630.

Section 2.2 ‘Methods’ – can you provide more details as to the how ‘behind the scenes’ insights were used and incorporated into the research? It is a little ambiguous at the moment. In addition, could you provide additional details as to how you linked key messages with the actions of promoters. Finally, can you provide reasoning for the selection of reports and articles. I imagine that you would argue that these are the most prominent and influential, but what criteria do you use? How do you account for the impact and/ or influence of other reports or academic publications? Why are these deemed less influential?  

Results: Do you incorporate any critical voice of the GND? You highlight the potential bias of drawing on the GNDs own visionaries, and highlight the use of more critical media publications and social theory?

Results: You have a compelling and interesting story to tell. To help the reader, I wonder if you could explore the idea of having a visual timeline? To track the development of the idea and key additions and moments?

 Discussion:

This raises good clear points and considers, for instance, the limitations of the research. It ends somewhat abruptly, I would consider adding a short conclusion section, to round off the article.

Specific comments:

Page 2, line 47 – can you give examples?

Page 2: you note “It therefore has important implications for TA, RRI and SR. Furthermore, TA, RRI and SR could potentially provide relevant expertise to make GND debates more reflexive and complex”… but have not to this point criticised the GND approach….

Pg. 2, line 80 – you note ‘this chapter’. I think this is a typo.

Author Response

Thank you for this constuctive and helpful review. We have added GND literature in the introduction and clarified our positioning in relation to it. The theory and methods section has been updated based on your feedback. The results section covers more critical perspectives to point at further research needs. The text includes a timeline of the vision in the making now. We have added dimensions of cirtique of the GND in the discussion chapter since the results chapter is meant for the analysis of the vision and its promoters. A conclusion chapter rounds off the article.

Thanks for the specific comments, which we have addressed.

Reviewer 2 Report

This article attempts to explain how the Green New Deal envisions a radical restructuring of science, technology, society, and the economy with the goals of ecological sustainability and social justice.  The paper does a good job of articulating the GND and what it envisions.  What is missing from this paper is an normative, ethical, moral, or scientific critique of the GND.  Is the GND realistic?  Most major transformations do not happen deliberately; they are usually the result of the actions of different players, driven by individual motivations and interests, market forces, etc. rather than overarching visions dictated by politics.  For example, consider transformations brought about by personal computers and the internet.  These did not happen by a top-down implementation of a grand vision or plan by governments or others.  Rather, a serious of choices led to a restructuring of society, economy, technology, science, etc. without any grand vision in mind.  These changes happened because of necessity and opportunity, rather than by political will.  Indeed, if one looks at attempts to bring about major transformations from the top-down, grand political vision approach, the results have been disastrous.  Cases in point: the Russian revolution and Stalinism; Nazi Germany; and China's cultural revolution.  What are the potential downfalls, problems, obstacles, resistance, unintended consequences of the GND?  These are not addressed in this paper.    

Author Response

Thank you for your review that acknowledges the main feat of the article – articulating the making of the GND vision. Furthermore, you raise an imporant point on the relationship of visions and complex social reality, i.e. on imagined and actual social reality. 

The normative claim of the article is that from the perspective of technology assessment, sustainability research and responsible research and innovation GND policies should be welcomed and be a field for constructive and critical engagement. However, we do not argue for top-down authoritarian management and most GND visions neither do so. 

We have realised, however, that we have not sufficiently discussed the relationship of visions and more complex processes of change that you mention in your review. We have clarified in the theory chapter that a) visions are but one dimension of transformations and that b) these transformations are not achievable by a grand plan or one steering agent alone. Yet, c) politics and state institutions play a role in shaping transformations and here the question is d) how is this being done. 

Therefore, our article does not seek to answer whether the GND is this realistic, but instead which social realities are shaped by GND visions.

To the introduction and discussion we have added critical perspectives on the GND which need further research, yet are beyond the scope of this article. 

Reviewer 3 Report

I think it is a valuable contribution regrading to actuality and approach. It nicely considers the TA, RRI and SR as potential "influencers" in the context of GND  development and discussion.

As such, it is already a good reading for any, who is already familiar with the subject.

But for the newcomer, I would expect a bit more analysis e.g. about RRI and GND (i.e. is it only engagement in the core of RRI or do the other elements matter too?; or how sustainability divided in the three dimensions is weighted in the GND approach(es)).

It seems also, that this is very "western" approach and discussion? A few words about global perspective regrading GND or SDGs (or statement of limitation of study) would help the reader to understand the context of analysis.

As we know, Green Deal is already established in the Horizon Europe funding structure (https://ec.europa.eu/info/research-and-innovation/strategy/strategy-2020-2024/environment-and-climate/european-green-deal_en). So, at least for me, the following sentence- Page 10), needs clarification. 

"This could start by aligning funding 
in the US and the EU with GND-inspired policies."

Author Response

Thank you for your review. With the “newcomer” in mind we have extended the discussion of RRI and sustainability in relation to the GND in the discussion chapter. It is beyond the articles scope, however, to analyse specific GND policies in relation to concepts of RRI or sustainability. The focus is on the foundational visionary narratives that have emerged and based on which policies are being shaped by now. 

We have acknowledged the eurocentrism in GND politics and pointed to the need for wider, inclusive and global perspectives. 

We have clarified the sentence regarding funding and GND policies. 

Round 2

Reviewer 1 Report

The authors have engaged with my comments to a good degree and substantially improved the paper. I now believe that it is ready for publication.

Reviewer 2 Report

Good job with revisions